# Diet-Induced Obesity in Mice Affects the Maternal Gut Microbiota and Immune Response in Mid-Pregnancy

**DOI:** 10.3390/ijms25169076

**Published:** 2024-08-21

**Authors:** Lieske Wekema, Sam Schoenmakers, Nicole Schenkelaars, Anne Laskewitz, Romy H. Huurman, Lei Liu, Lisa Walters, Hermie J. M. Harmsen, Régine P. M. Steegers-Theunissen, Marijke M. Faas

**Affiliations:** 1Department of Pathology and Medical Biology, University Medical Center Groningen, University of Groningen, Hanzeplein 1, 9713 GZ Groningen, The Netherlands; a.laskewitz@umcg.nl (A.L.); r.h.huurman@umcg.nl (R.H.H.); 2Department of Obstetrics and Gynaecology, Erasmus Medical Center, Dr. Molewaterplein 40, 3015 GD Rotterdam, The Netherlands; s.schoenmakers@erasmusmc.nl (S.S.); n.schenkelaars@erasmusmc.nl (N.S.); r.steegers@erasmusmc.nl (R.P.M.S.-T.); 3Department of Medical Microbiology, University Medical Center Groningen, University of Groningen, Hanzeplein 1, 9713 GZ Groningen, The Netherlands; l.liu@umcg.nl (L.L.); l.walters@umcg.nl (L.W.); h.j.m.harmsen@umcg.nl (H.J.M.H.); 4Department of Obstetrics and Gynaecology, University Medical Center Groningen, University of Groningen, Hanzeplein 1, 9713 GZ Groningen, The Netherlands

**Keywords:** maternal obesity, gut microbiota, immune response, mid-pregnancy, murine model

## Abstract

Maternal obesity during pregnancy is associated with adverse pregnancy outcomes. This might be due to undesired obesity-induced changes in the maternal gut microbiota and related changes in the maternal immune adaptations during pregnancy. The current study examines how obesity affects gut microbiota and immunity in pregnant obese and lean mice during mid-pregnancy (gestational day 12 (GD12)). C57BL/6 mice were fed a high-fat diet or low-fat diet from 8 weeks before mating and during pregnancy. At GD12, we analyzed the gut microbiota composition in the feces and immune responses in the intestine (Peyer’s patches, mesenteric lymph nodes) and the peripheral circulation (spleen and peripheral blood). Maternal obesity reduced beneficial bacteria (e.g., *Bifidobacterium* and *Akkermansia*) and changed intestinal and peripheral immune responses (e.g., dendritic cells, Th1/Th2/Th17/Treg axis, monocytes). Numerous correlations were found between obesity-associated bacterial genera and intestinal/peripheral immune anomalies. This study shows that maternal obesity impacts the abundance of specific bacterial gut genera as compared to lean mice and deranges maternal intestinal immune responses that subsequently change peripheral maternal immune responses in mid-pregnancy. Our findings underscore the opportunities for early intervention strategies targeting maternal obesity, ideally starting in the periconceptional period, to mitigate these obesity-related pregnancy effects.

## 1. Introduction

Maternal obesity during pregnancy is associated with adverse pregnancy outcomes such as miscarriage, gestational diabetes mellitus, preeclampsia, and an increased risk for congenital malformations [1,2,3]. Increasing evidence indicates that maternal obesity also enhances the risk of childhood obesity due to the prolonged effects of macrosomia and metabolic syndrome in the offspring [4,5]. With the increasing prevalence of obesity in women of reproductive age, maternal obesity has become a major public health concern [6]. However, despite these observations, the mechanisms by which obesity impacts pregnancy outcomes are still largely unknown [7]. Insight into these mechanisms and time points at which obesity induces adverse pregnancy outcomes can lead to better strategies for intervention.

During an uncomplicated pregnancy, adaptations of the maternal immune response, both locally at the maternal–fetal interface as well as in the peripheral circulation, are necessary to prevent the rejection of the semi-allogeneic fetus and to create a favorable intrauterine environment [8]. These adaptions are highly dynamic, vary at different gestational stages, depending on the anatomical location, and include, for instance, adaptations in the Th1/Th2/Th17/Treg balance as well as monocyte subsets [9,10,11,12,13]. Derangements in maternal immune adaptations in the peripheral circulation are associated with, for example, preeclampsia, miscarriage, fetal growth restriction, and premature birth [12,14,15,16,17,18,19]. In addition, studies have shown that maternal obesity during pregnancy is associated with immunological derangements. We and others have shown that maternal obesity causes disruption of the intestinal and peripheral Th1/Th2/Th17/Treg axes at the end of pregnancy in humans and mice, skewing the Th profile towards Th1 and Th17 [20,21]. Such derangements might thus be responsible for the adverse pregnancy outcomes in obese pregnancies [20,22,23,24]. 

Immune responses in general are, at least partly, influenced by the gut microbiota [25]. How this process is orchestrated during pregnancy is currently still largely unknown. Bacteria can directly interact with gut epithelial and/or intestinal immune cells, such as dendritic cells (DCs) in the Peyer’s patches (PPs), to activate intestinal immune cells, which can subsequently travel to the peripheral circulation via the mesenteric lymph nodes (MLNs) to alter peripheral immunity [26,27]. Also, gut microbes produce a variety of microbial metabolites, including short-chain fatty acids (SCFAs), that are involved in immunomodulation [28]. Throughout an uncomplicated pregnancy, the composition and structure of the gut microbiota are profoundly altered [29,30,31]. It is thought that these microbial changes support beneficial obstetrical outcomes with long-term health benefits for the offspring, for example, by adapting the maternal immune response [15,21,32,33,34]. Pregnant women with obesity are known to have a different composition and structure of the gut microbiota compared to those with normal weight, which can accordingly affect their immune responses [35,36,37,38].

In a recent study, we have shown that high-fat diet-induced obesity in mice altered their gut microbiota before and during pregnancy. This change affected maternal peripheral and intestinal immune responses at the end of pregnancy, which was associated with decreased fetal weight [21]. In the present study, we focus on mid-pregnancy in the mouse and investigate whether obesity-induced gut dysbiosis during pregnancy is associated with changes in the intestinal and peripheral immune response at gestational day 12 (GD12) as compared to lean pregnant mice.

## 2. Results

### 2.1. Maternal Obesity Does Not Affect Pregnancy Outcome at Mid-Pregnancy

Mean fetal weight, placental weight, total number of fetuses, and the percentage of alive fetuses per dam were used to assess pregnancy outcome and were measured at gestational day 12 (GD12) (Figure 1). 

Although a slightly lower fetal weight was observed in offspring exposed to HFD-induced maternal obesity, there were no statistically significant differences in fetal weight, placental weight, total fetal number, or the percentage of alive fetuses between the obese and lean group.

### 2.2. Maternal Obesity Alters the Gut Microbiota Composition at Mid-Pregnancy

Since maternal obesity is linked to microbial gut dysbiosis, we examined whether the gut microbiota composition of obese mice differed from that of lean mice at GD12. Fecal pellets were analyzed for microbiota diversity and relative abundance using 16S rRNA sequencing. 

PCA analysis revealed that the gut microbiota of obese mice significantly differed from that of lean mice at GD12 (PERMANOVA, *p* < 0.01) (Figure 2); however, no differences in alpha diversity were observed, as indicated by the Shannon index (Appendix A). Subsequently, differences in bacterial phyla and genera were compared between obese and lean mice. On the phylum level (Figure 3), we found a decreased abundance of *Actinobacteria* (*p* < 0.01) and *Verrucomicrobia* (*p* < 0.01) in obese compared to lean mice. On the genus level (Figure 4), we found decreased abundances of *Bifidobacterium* (*p* < 0.01), *Olsenella* (*p* < 0.05), *Muribaculum* (*p* < 0.01), *Parabacteroides* (*p* < 0.05), *Christensenella* (*p* < 0.05), *Ruthenibacterium* (*p* < 0.05), *Faecalibaculum* (*p* < 0.05), *Turicibacter* (*p* < 0.05), *Parasutterella* (*p* < 0.05), and *Akkermansia* (*p* < 0.01) in obese versus lean mice, whilst increased abundances of *Adlercreutzia* (*p* < 0.01), *Enterococcus* (*p* < 0.01), *Lactococcus* (*p* < 0.001), *Acetatifactor* (*p* < 0.05), *Schaedlerella* (*p* < 0.05), *Amedibacillus* (*p* < 0.01), and *Clostridium XVIII* (*p* < 0.05) were found in obese mice.

### 2.3. Maternal Obesity Alters the Intestinal Immune Response at Mid-Pregnancy

As the gut microbiota are involved in the immunomodulation of the intestinal immune system, we quantified T helper (Th) cell subsets (Figure 5) and dendritic cell (DC) subsets (Figure 6) in the Peyer’s patches (PP) and mesenteric lymph nodes (MLN) of obese and lean mice. An overview of these immune cells and their function can be found in Appendix A. 

In obese mice, the percentage of Th17 cells (*p* < 0.05) was increased in the PPs, whereas the percentage of Th1 cells (*p* < 0.05) was decreased in the MLNs. No differences were found for the percentages of Th1, Th2, and Treg cells in the PPs or percentages of Th2, Th17, and Treg cells in the MLNs. Only in the PPs of obese mice, the percentage of DCs was decreased (*p* <0.01). No differences between the two groups were observed in the percentages of CD103^+^CD11b^−^, CD103^+^CD11b^+^, or CD103^−^CD11b^+^ DCs subsets in either of these tissues.

### 2.4. Obesity-Induced Changes in the Maternal Intestinal Immune Response Correlate with Obesity-Induced Changes in Maternal Gut Microbiota 

To understand the relationship between the immunological changes in the intestine and the shifts in gut microbiota genera between obese and lean mice, we correlated individual immune cell data and individual microbiota abundances on the genus level of the same mouse. Spearman’s rank correlation coefficients are displayed in a heatmap (Figure 7). Moreover, since mesenteric lymph nodes (MLN) drain lymph from various locations of the gastrointestinal tract including the Peyer’s patches (PPs), we also correlated individual immune cell data in the PPs and MLNs of the same mouse (Appendix A). 

The following significant correlations between the obesity-associated gut microbiota and immune cell subsets were identified. Most of the genera, which were increased in obese mice as compared to lean mice, correlated negatively with the percentage of DCs in the PPs. Several bacterial genera, including *Amedibacillus*, *Adlercreutzia*, and *Acetatifactor*, which were increased in obese mice, correlated positively with the percentage of Th17 cells in the PPs. The bacterial genera that showed decreased abundance in obese mice compared to lean mice exhibited contrasting effects. Several genera, including *Olsenella*, *Turicibacter*, *Bifidobacterium*, and *Akkermansia* correlated positively with the percentage of dendritic cells in the PPs. The genera *Muribaculum* and *Turicibacter* correlated positively with the percentage of Th1 cells in the MLNs, whilst the genus *Faecalibaculum* correlated negatively with the percentage of Th17 cells in the PPs.

### 2.5. Maternal Obesity Alters the Peripheral Immune Response at Mid-Pregnancy

We next investigated whether changes between obese and lean pregnant mice were present in the peripheral immune system. To this end, we quantified splenic T helper (Th) cell subsets and the capacity of splenic Th cells to produce IFN-γ, IL-4, IL-17A, and IL-10 upon ex vivo stimulation with PMA (Figure 8). In addition, we also determined monocyte subsets and activation status, indicated by CD80 expression, in maternal blood (Figure 9). An overview of these immune cells and their function is provided in Appendix A.

In obese mice, percentages of splenic Th2 (*p* < 0.05) and Treg (*p* < 0.01) cells were increased as compared to lean mice, but no differences were found for the percentages of Th1 and Th17 cells. We also found that the percentages of IL-17A (*p* < 0.01) and IL-10 (*p* < 0.05) producing splenic T helper cells were, respectively, decreased and increased in obese mice. No differences were observed in the percentages of IFN-γ- and IL-4-producing T helper cells upon ex vivo stimulation.

In the blood of obese mice, classical monocytes were increased (*p* < 0.05), whilst non-classical monocytes were decreased (*p* < 0.05). No differences were observed for CD80 expression on classical, intermediate, or non-classical monocytes between obese and lean mice.

### 2.6. Obesity-Induced Changes in the Maternal Peripheral Immune Response Correlate with Obesity-Induced Changes in Maternal Gut Microbiota

To understand the relationship between the observed immunological changes in the periphery and changes in gut microbiota genera, individual peripheral immune cell data, that were different between obese and lean pregnant mice, and individual microbiota abundance on the genus level, that were different between obese and lean pregnant mice, of the same mouse were correlated. Spearman’s rank correlation coefficients are presented in heatmaps (Figure 10).

Several significant correlations between the obesity-associated gut microbiota and peripheral immune cell subsets were identified. Most of the bacterial genera, which were increased in obese mice as compared to lean mice, positively correlated with percentages of splenic Treg cells and negatively correlated with IL-17A production by splenic Th cells. Only certain genera that were elevated in obese mice, such as *Adlercreutzia*, *Clostridium XVIII*, *Acetatifactor*, *Lactococcus*, and *Amedibacillus* correlated positively with the percentage of classical monocytes, Th2 cells, and/or IL-10 production by splenic Th cells and/or negatively correlated with non-classical monocytes. Opposite effects were observed for the bacterial genera which were decreased in obese mice compared to lean mice. Most of these genera negatively correlated with percentages of Treg, Th2, and IL-10 production by splenic T helper cells and positively correlated with IL-17A production by splenic T helper cells. Several specific genera that were decreased in obese versus lean mice, including *Ruthenibacterium*, *Muribaculum*, and *Bifidobacterium*, correlated negatively with classical monocytes and/or positively with non-classical monocytes.

### 2.7. Intestinal Immune Cells in the PPs and MLNs Correlate with Peripheral Immune Cells in the Spleen and Blood 

Intestinal immune responses can regulate peripheral immune responses. Therefore, we correlated individual immune cell data in the Peyer’s patches (PPs) and spleen/blood and individual immune cell data in mesenteric lymph nodes (MLNs) and spleen/blood. Spearman’s rank correlation coefficients are presented in heatmaps (Figure 11).

Significant correlations were observed between immune cell subsets in the PPs and peripheral immune cells (Figure 11A). The percentage of Th2 cells in the PPs correlated negatively with IL-17A production by splenic Th cells. The Th17 cell percentage in the PPs correlated positively with Th1 cells in the spleen, whereas the percentage of Treg cells in the PPs negatively correlated with CD80 expression on classical monocytes in the blood. DCs in the PPs were positively and negatively correlated with, respectively, IL-17A and IL-10 production by splenic T helper cells. 

Significant correlations were also found between immune cell subsets in the MLNs and peripheral immune cells (Figure 11B). The percentage of Th1 cells in the MLNs negatively correlated with the percentage of splenic Treg cells. The Th2 cell percentage in the MLNs correlated negatively with the percentage of splenic Th1 cells and IL-4 production by splenic Th cells. Th17 cells in the MLNs correlated negatively with IL-4 production by splenic Th cells. CD103^+^CD11b^−^ DCs negatively correlated with non-classical monocytes in the blood. CD103^−^CD11b^+^ DCs in the MLNs were positively and negatively correlated with, respectively, intermediate monocytes and CD80 expression on non-classical monocytes.

## 3. Discussion

Our findings indicate that during mid-pregnancy, mice with obesity induced by a high-fat diet (HFD) before and during pregnancy exhibit a lower abundance of specific bacterial genera known for their beneficial effects on maternal and fetal health, such as *Bifidobacterium* and *Akkermansia* [39,40,41,42,43,44,45,46]. Maternal obesity also affected the immune response during mid-pregnancy, as we found derangements in the Th1/Th2/Th17/Treg axis, monocyte subsets, and dendritic cells. Our results also showed many correlations between obesity-associated bacterial genera and intestinal/peripheral immune anomalies and between the intestinal and peripheral immune responses. This confirms our hypothesis that maternal obesity deranges both particular bacterial genera in the gut and the maternal intestinal immune system, subsequently impacting the maternal peripheral immune system, as early as mid-pregnancy. An overview of all the HFD-induced obesity-associated changes found in this study is depicted in Figure 12.

Previously, we and others have shown that maternal obesity, induced by an HFD before and during pregnancy, decreases fetal weight at the end of murine pregnancy (gestational day 18 (GD18)) [21,47,48,49,50]. Although not statistically significant, fetal weight already appeared to be lower in the obese animals at GD12 of pregnancy as compared to their lean counterparts. This suggests that fetal growth in obese mice is already slightly disturbed at day 12 of pregnancy. Since the first half of pregnancy in mice mainly involves fetal and placental development, and fetal growth in mice predominantly occurs from E14 until the end of pregnancy, this could explain the lack of detectable effect on fetal growth or placental weight [51]. Moreover, several studies show that placental vascularization and growth disturbances during early development may lead to adverse pregnancy outcomes beyond this period [21,51,52]. 

The primary sites for interaction between gut microbiota and the intestinal immune system are the Peyer’s patches (PPs). We show clear differences in the frequency of different immune populations in the PPs. Obesity was associated with increased Th17 cells, which is in alignment with our previous study in which we observed sustained increased Th17 cell-frequencies in the PPs at the end of pregnancy (GD18) [21]. This upregulation may be a response to changes in the obesity-associated gut microbiota, given that the microbiota influences the regulation and activity of Th17 cells in the gut [53]. Th17 cells are considered immune cells with both pathogenic and protective functions [54,55]. By the release of various cytokines such as IL-10 and IL-22, intestinal Th17 cells strengthen the intestinal barrier function and support the adhesion of commensals in the lumen [56,57,58]. Therefore, the upregulation of Th17 cells in the PPs of obese pregnant mice could be a counter-regulatory response to the observed gut dysbiosis. In addition, we also found a decreased percentage of Th1 cells in the mesenteric lymph nodes (MLNs) of obese versus lean pregnant mice. The fact that Th1 cells in the MLN negatively correlated with CD103^+^CD11b^+^ DCs in the PPs, suggests that these dendritic cells may have induced the decrease in Th1 cells. These findings are in line with the previously mentioned findings of increased Th17 cells, as CD103^+^CD11b^+^ DCs are unique to the intestine and are essential for the maintenance of mucosal immunity and barrier function and play a key role in the differentiation of T cells [59,60]. 

The intestinal microbial and immunological anomalies that were induced by the obese state also seem to affect immunological changes within the periphery due to observed correlations between immune cell subsets in the PPs/MLNs and in the spleen/blood. The spleen of obese mice showed increased percentages of Th2 and Treg cells, while classical monocytes were increased, and non-classical monocytes decreased in the blood. Based upon the cytokine production patterns of splenic T helper cells upon stimulation with PMA, obesity seems to enhance the capacity of these cells to produce IL-10, while it diminished IL-17A production. In combination, the observed changes in the periphery, especially the increase in Treg and IL-10 and the decrease in IL-17A production indicate a more anti-inflammatory immune response in obese pregnancy in the mouse at GD12 [9,61]. This is in contrast to our previous mouse study, which showed that maternal obesity skewed the Th profile towards Th1 and Th17 at GD18 [21]. These findings suggest that maternal immune adaptations in mice are regulated differently at different gestational stages during obese pregnancy as compared to lean pregnancy. This stage-specific regulation of the immune response in obese pregnancy could be related to the highly dynamic nature of the immune response that is observed during human pregnancy [11,62,63]. At the start of pregnancy, a Th1-type immune response and inflammation are needed for proper blastocyst implantation and placental development [11]. Following implantation, the immune response shifts towards a Th2-type profile, which is crucial for fetal tolerance [11]. The shift towards Th2 and Treg at GD12 in obese pregnancy may suggest an earlier-than-normal transition to Th2, which may hamper proper placental development. Therefore, the disturbed maternal immune adaptations seen at GD12 in obese pregnancy are likely the basis for the adverse pregnancy outcomes observed in late murine pregnancy [21].

Specific bacterial genera known for beneficial health effects, such as those producing short-chain fatty acids (SCFAs), were found to be reduced in obese mice. These included *Bifidobacterium*, *Olsenella*, *Muribaculum*, *Parabacteroides*, *Christensenella*, *Ruthenibacterium*, *Faecalibaculum*, *Turicibacter*, *Parasutterella*, and *Akkermansia*. Obesity also increased certain genera like *Adlercreutzia*, *Enterococcus*, *Lactococcus*, *Acetatifactor*, *Schaedlerella*, *Amedibacillus,* and *Clostridium XVIII*. Our results align with observations in pregnant women with obesity, who also show reduced levels of *Bifidobacterium*, *Olsenella*, *Christensenella*, and *Akkermansia* and increased levels of *Adlercreutzia*, *Enterococcus*, and *Clostridium* in the gut [36,37,64,65,66,67]. These similarities suggest that humans may experience comparable microbiota–immune interactions. This warrants further investigation into how these associations affect pregnancy. Future studies should examine the causal relationships between these bacterial genera and immune modulation, as well as explore potential therapeutic interventions targeting gut microbiota to improve pregnancy outcomes in individuals with obesity.

The shifts in gut microbiota composition were correlated with the immunological changes that we observed in obese mice, suggesting that they play a role in the immunological changes we observed in obese pregnant mice. In the intestinal immune response, several genera that were increased or decreased in obese mice, positively or negatively correlated with Th17 and dendritic cell percentages in the PPs, suggesting a role for these bacterial genera in regulating PP immune cells. This is in line with various studies, showing that the microbiota is important for regulating Th17 responses [25,26,53]. The same genera also correlated with peripheral immune cells, such as Tregs and Th2 cells. Recently, *Adlercreutzia* species have been shown to convert bile acids into 3-oxolithocholic acid (3-oxoLCA) [68]. This metabolite inhibits Th17 cells [68]. Indeed, in our study, increased *Adlercreutzia* species in the gut microbiota correlated with decreased IL-17A production after stimulation of peripheral T cells. Interestingly, increased *Adlercreutzia* species positively correlated with Th17 cells in the PP, suggesting a different effect of these species on Th17 cells in the PPs. It is also especially interesting to note the effects of *Bifidobacterium*, *Christensenella,* and *Akkermansia* species, since several of these species, when administered as probiotics, have already proven to exert anti-obesogenic effects in (pregnant) rodents and/or humans [40,41,42,43,44,45,46,69,70,71]. 

*Bifidobacterium* species, important producers of SCFAs, are able to modulate the immune system [72,73] and impact pregnancy by supporting maternal body adaptations, placental morphogenesis, nutrient transport, fetal metabolism, and growth in mice [39]. Supplementation with *Bifidobacterium* as probiotics during pregnancy also lowers serum levels of high-sensitivity C-reactive protein, reduces total HDL and LDL cholesterol as well as triglyceride levels, and improves glucose tolerance in (obese) women [40]. These effects likely reduce the risks of adverse pregnancy outcomes. Bacterial species of the genus *Akkermansia* are well-known mucin-degrading bacteria that generally strengthen the integrity of the intestinal barrier [74]. One study described that *Akkermansia muciniphila* supplementation during pregnancy enhanced intestinal barrier function in conventional mice [75]. Therefore, the decreased abundance of *Akkermansia* in obese pregnancies, as observed in our study, may contribute to a weakened intestinal barrier in these mice. This aligns with our explanation that the percentage of Th17 cells in the PPs increases as a compensatory reaction to obesity-induced gut dysbiosis, as these cells are involved in improving intestinal barrier function [57,58]. Our data thus suggest that *Bifidobacterium* species and *Akkermansia* species may be used as probiotics in obese pregnancy to improve the maternal immune response.

Our study has some limitations that should be considered. First, while our correlation analysis identifies associations between variables, it does not establish causation, limiting our ability to draw direct cause-and-effect conclusions from the data. To address this, further studies are underway in which feces from obese and lean pregnant mice will be transplanted into germ-free mice to study the causal effects of the microbiota on immune responses. Additionally, we used only one mouse strain in our experiments, which may limit the extent to which our findings can be applied to other genetic backgrounds or strains that might show different responses. In our study, we used syngeneic pregnancies to exclude the effect of semi-allogeneity on the immune response, allowing us to evaluate the effects of the microbiota more clearly. Future studies should include semi-allogeneic pregnancies to verify whether similar microbiome effects on the maternal immune response occur in these mice. We did not investigate the impact of the diet per se on gut microbiota, which could influence microbiota composition and study outcomes; future research should explore this aspect to gain a more comprehensive understanding. Lastly, the findings from our mouse model may not directly translate to human conditions due to inherent species differences. Further studies involving human subjects are needed to validate the relevance and applicability of our results.

Concluding remarks: Our data demonstrate that HFD-induced maternal obesity can profoundly affect both the maternal gut microbiota composition and intestinal/peripheral immune responses already at mid-pregnancy in C57BL/6 mice. Although fetal weight was not significantly affected at this stage, the observed anomalies are probably the basis of adverse pregnancy outcomes that are observed during late murine pregnancy. Our findings are essential for the timing of possible intervention strategies to prevent the adverse effects of maternal obesity on both the mother and the fetus. As anomalies in both gut microbiota composition and immunity already develop in the first period of pregnancy and given the significant influence of both microbiota and immune system dynamics on implantation and subsequent fetal development, intervention should preferably already start before or at least during the periconceptional period [11,76,77,78]. 

## 4. Materials and Methods

### 4.1. Animals

C57BL/6JOlaHsd mice, both female and male, were obtained from Envigo in the Netherlands when they were 6 weeks old. Upon arrival, the female mice (*n* = 16) were sorted into groups based on body weight and were then assigned to either a high-fat diet (HFD; 60 kcal% fat, D124921i, Research Diets Inc., New Brunswick, NJ, USA) or a low-fat diet (LFD; 10 kcal% fat, D12450Ji, Research Diets Inc.) for 8 weeks, resulting in obese and lean mice, respectively. Mice were classified as obese if their body weight was 30% higher than the lean group (Appendix A). Female mice were housed together in ventilated cages in groups of 3-5, separated into obese and lean categories, with a 12/12 h light/dark cycle and unlimited access to food and water. Male mice were housed individually under similar conditions and were given the LFD. After the induction of obesity, vaginal smears were collected from the females, and those in proestrus were paired with a male overnight. During this mating period, males were fed the same diet as their female partners. The following morning, the presence of a vaginal plug was checked to confirm mating, marking that day as E0 of pregnancy. The body weight of pregnant females was monitored throughout gestation (Appendix A). On gestational day 12 (GD12) of pregnancy, females were euthanized through exsanguination via cardiac puncture under general anesthesia (isoflurane) (*n* = 8 for each group: obese and lean). Blood samples were collected in EDTA tubes (BD Biosciences, Breda, The Netherlands), and spleens, mesenteric lymph nodes (MLN), and Peyer’s patches (PP) were harvested and kept on ice in RPMI medium (Thermo Fisher Scientific, Waltham, MA, USA) with 10% decomplemented fetal calf serum (dFCS; Thermo Fisher Scientific) until cell isolation within 2 h. PPs and MLNs were analyzed to study the intestinal immune response, while blood and spleen samples were used to evaluate the peripheral immune response. Fecal samples were collected from the colon, frozen in liquid nitrogen, and stored at −80 °C for subsequent microbiota analysis. Pregnancy outcomes were measured by determining fetal weight, placental weight, the number of fetuses, and the percentage of live fetuses on GD12. The animal experiments in this study were approved by the Central Committee on Animal Experimentation in the Netherlands (CCD application number AVD10500202010704).

### 4.2. Gut Microbiota Composition

DNA isolation, PCR, 16S rRNA gene sequencing, quality control, and taxonomy assignment were performed according to the methods documented previously [21,79]. In brief, cell lysis was performed with a tissue homogenizer in combination with lysis buffer (500 mM NaCl, 50 mM Tris-HCl [pH 8.0], 50 mM EDTA, and 4% (*v*/*v*) SDS). Proteins were removed with ammonium acetate (10 M), nucleic acids were precipitated with isopropanol, and RNA was removed by RNase (500 µg/mL, cat #11579681001, Roche, Basel, Switzerland). DNA purification was performed with the QIAmp DNA Mini Kit (Qiagen, Hilden, Germany). The V3–V4 region of the 16S rRNA gene was amplified with modified barcoded 341F and 806R primers. Amplicons were sequenced with a Miseq Illumina sequencing platform. Paired-end reads, demultiplexed based on barcode, were retrieved from the Illumina platform and were instructed by EasyAmplicon analysis pipeline. Joined-reads were quality-controlled with maximum error rate of 1%, and primer sequences were cut with VSEARCH. Denoising was performed with USEARCH and VSEARCH. Amplicon Sequence Variants were assigned based on Ribosomal Database Project set 18.

### 4.3. Cell Isolation from Tissues

To obtain single-cell suspensions from the spleens, MLNs, and PPs, tissues were disrupted between two object glasses and filtered using tubes with strainer caps as documented before [21]. Ammonium chloride (0.16 M NH_4_Cl, 0.01M KHCO_3_, 0.03 mM CH_3_COONa-2H_2_O) was used to lyse red blood cells in the splenocyte suspension. Cell suspensions were used for cell staining as described below. Excess splenocytes were preserved by freezing them in a solution of 10% DMSO (in dFCS) and were stored at −80 °C until cytokine production by stimulated splenic Th cells could be measured.

### 4.4. T Helper Cell Staining 

Cells from the spleen, MLNs, and PPs were stained for Th subsets according to the protocol described in our previous paper [21]. In brief, live/dead staining was performed using Zombie NIR solution (1:1000, cat #423105, Biolegend, San Diego, CA, USA) for 30 min. Extracellular blocking medium (78% (*v*/*v*) FACS buffer (2% (*v*/*v*) dFCS in 1x DPBS (Thermo Fisher Scientific)), 20% (*v*/*v*) rat serum (Jackson, UK), and 2% (*v*/*v*) purified anti-mouse CD16/32 (cat #101302, Biolegend)) was used to prevent non-specific binding of antibodies. Cells were stained for extracellular markers with monoclonal antibodies recognizing CD3, CD4, and CD8 (Appendix A). Cells were fixed with FACS lysing solution (1x, BD Biosciences) and were permeabilized with permeabilization buffer (1x, eBioscience, Vienna, Austria). After incubation with intracellular blocking medium (80% (*v*/*v*) permeabilization buffer and 20% (*v*/*v*) rat serum), cells were stained for intracellular makers with monoclonal antibodies recognizing Tbet (Th1), Gata-3 (Th2), RORγT (Th17), and FoxP3 (Treg) (Appendix A).

### 4.5. Dendritic Cell Staining 

Cells from the MLNs and PPs were stained to identify dendritic cell (DC) subsets. Each sample, containing 1 × 10^6^ cells, was added to a 96-well plate (Corning, Amsterdam, The Netherlands). Following centrifugation, the cells were washed twice with 200 μL of DPBS. The cells were then resuspended in 100 μL of Zombie Green solution (1:1000, cat #423111, Biolegend) and incubated for 30 min at room temperature. After another round of centrifugation, the cells were washed twice with 200 μL of FACS buffer. The cells were then resuspended and incubated in 50 μL of extracellular blocking medium for 10 min. Following this, the cells were centrifuged, resuspended, and incubated with 25 μL of extracellular antibody mix, which included monoclonal antibodies against MHCII, CD11c, CD64, CD19, B220, Nkp46, CD103, and CD11b (Appendix A) for 30 min. The cells were then washed twice in 150 μL and 200 μL of FACS buffer, respectively, and fixed in 200 μL of FACS lysing solution for 30 min. Subsequently, the cells were washed three times with 200 μL of FACS buffer and stored in 200 μL of FACS buffer at 4 °C. Data acquisition was performed within 24 h. During washing, centrifugation was performed at 600× *g* for 3 min at 4 °C. All steps, post the addition of the viability solution, were carried out in the dark and on ice unless stated otherwise.

### 4.6. Cytokine Production by Splenic T Helper Cells 

Splenic Th cells were stimulated ex vivo to assess cytokine production according to the protocol described in our previous paper [21]. In brief, cells were stimulated with phorbol 12-myristate 13-acetate (PMA, 20 ng/mL, Sigma-Aldrich, St. Louis, MO, USA) and ionomycin (1.8 µg/mL, Sigma-Aldrich) for 4 h at 37 °C in the presence of Brefeldin A (10 µg/mL, Sigma-Aldrich). Live/dead staining was performed using Zombie NIR solution. Extracellular blocking medium was used to prevent non-specific binding of antibodies. Cells were stained for extracellular markers with monoclonal antibodies recognizing CD3, CD4, and CD8 (Appendix A). Cells were fixed with FACS lysing solution and were permeabilized with permeabilization buffer. After incubation with intracellular blocking medium, cells were stained for intracellular cytokines with monoclonal antibodies recognizing IFN-γ, IL-4, IL-17A, and IL-10 (Appendix A).

### 4.7. Monocyte Staining 

Maternal EDTA blood was stained for monocyte subsets and activation status according to the protocol described in our previous paper [21]. In brief, extracellular blocking medium (78% FACS-EDTA buffer (5% (*v*/*v*) dFCS and 372 µg/mL EDTA (Sigma) in DPBS), 20% (*v*/*v*) rat serum, and 2% (*v*/*v*) purified anti-mouse CD16/32) was used to prevent non-specific binding of antibodies. Cells were stained for extracellular markers with monoclonal antibodies against CD11b, Ly6G, CD115, CD43, Ly6C, and CD80 (Appendix A). Cells were fixed and red blood cells were lysed with FACS lysing solution.

### 4.8. Flow Cytometry 

Data acquisition was conducted on a NovoCyte Quanteon Flow Cytometer (Agilent, Santa Clara, CA, USA) utilizing NovoExpress software version 1.6.2 (Agilent). Data analysis was carried out using FCS Express software version 6 (De Novo Software, Pasadena, CA, USA). Gating strategies for Th cells, cytokine production by splenic Th cells, and monocytes were performed as documented previously [21]. The gating strategy for DC subsets was performed as illustrated in Appendix A. Technical difficulties during processes such as immune cell staining and flow cytometry prevented us from obtaining data for all immune cells in every tissue for all mice. A summary of all immune cells stained in this study, along with their functions, is provided in Appendix A [79,80,81,82,83].

### 4.9. Statistical Analysis 

The composition of maternal gut microbiota was statistically analyzed using Past4 hammer [84]. Differences between groups were assessed with the PERMANOVA test. Prism software version 10 (GraphPad Software, San Diego, CA, USA) was used to determine statistical differences in bacterial phyla and bacterial genera and in Shannon index between the groups by means of the Mann–Whitney U test. 

Statistical analysis on pregnancy outcome (mean fetal weight, placental weight, fetal number, and the percentage of alive fetuses per dam), maternal immune cell subsets, maternal body weight, and maternal weight gain during pregnancy was performed using Prism software version 10. Initially, outliers were removed by means of the ROUT test (Q = 1%) (only for immune cell data). For pregnancy outcome and immune cells, the Mann–Whitney U test was used to determine differences between groups (obese versus lean). For maternal body weight and weight gain during pregnancy, the Kruskal–Wallis test followed by Dunn’s multiple comparisons test was used to evaluate whether there were differences between pregnancy days.

For evaluating correlations between bacterial genera and immune cell subsets and between immune cell subsets, SPSS was utilized to determine Spearman’s rank correlation coefficients. Heatmaps were produced using CLUSTVIS [85]. Euclidean distance and Ward’s linkage were used for cluster analysis. 

Data were considered significantly different when *p* < 0.05.

## Figures and Tables

**Figure 1 ijms-25-09076-f001:**
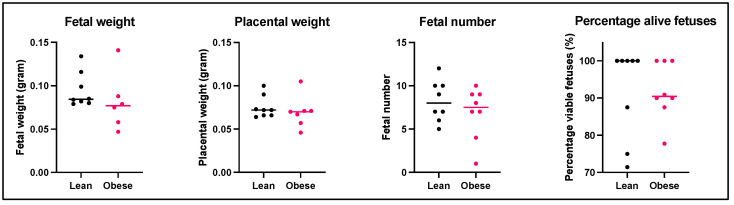
Mean fetal weight, placental weight, total fetal number, and percentage of alive fetuses per dam obtained from lean and obese mice. Data are displayed as individual values along with the median. Mann–Whitney U test. Lean mice: *n* = 8. Obese mice: *n* = 6–8.

**Figure 2 ijms-25-09076-f002:**
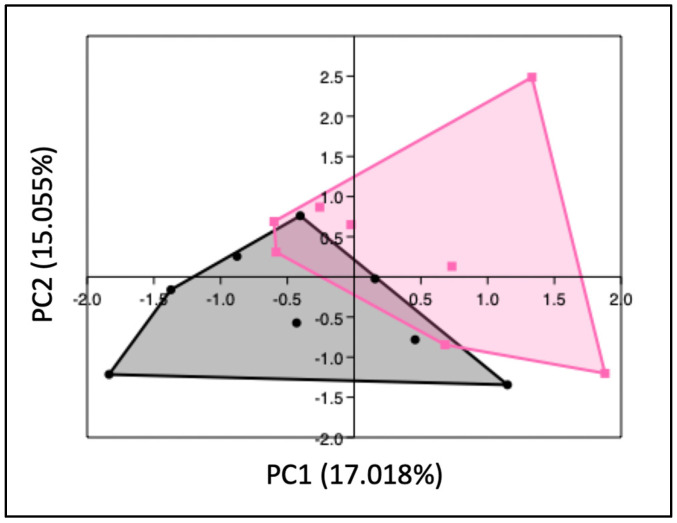
PCA plot showing the gut microbiota composition of lean (black dots) and obese (pink squares) mice at day 12 of pregnancy. Eigenvalues are shown at the axes, which represent the total amount of variance per principal component. Lean mice: *n* = 8. Obese mice: *n* = 8.

**Figure 3 ijms-25-09076-f003:**
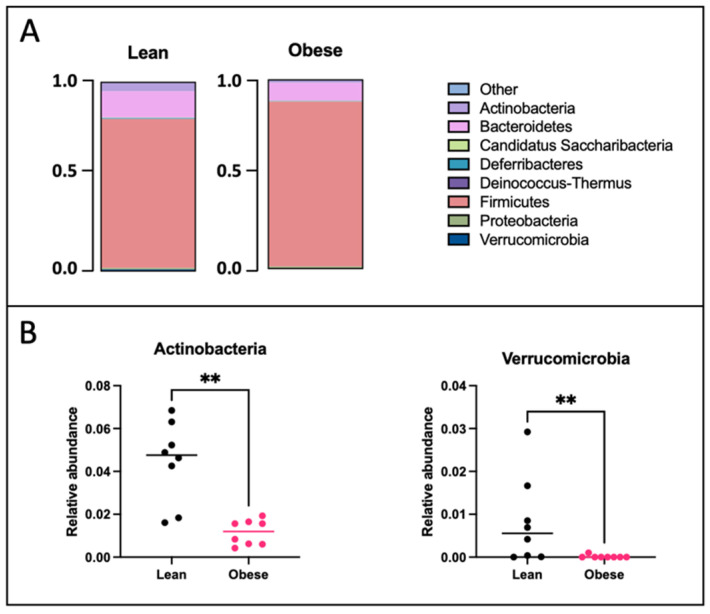
(**A**) Relative abundances of bacterial phyla in lean and obese mice at day 12 of pregnancy plotted in a bar chart. Lean mice: *n* = 8. Obese mice: *n* = 8. (**B**) Significantly different bacterial phyla between obese and lean mice at day 12 of pregnancy. Data are displayed as individual values along with the median. Mann–Whitney U test. ** *p* < 0.01. Lean mice: *n* = 8. Obese mice: *n* = 8.

**Figure 4 ijms-25-09076-f004:**
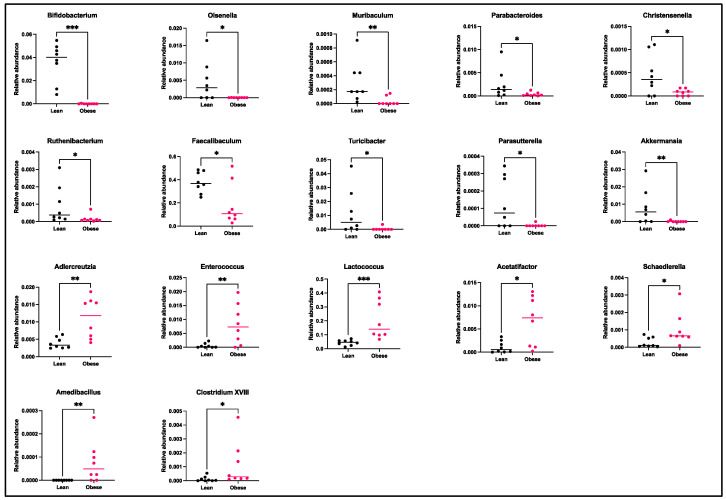
Relative abundances of the bacterial genera, which significantly differed between obese and lean mice at day 12 of pregnancy. Data are displayed as individual values along with the median. Mann–Whitney U test. * *p* < 0.05, ** *p* < 0.01, *** *p* < 0.001. Lean mice: *n* = 8 and obese mice: *n* = 8.

**Figure 5 ijms-25-09076-f005:**
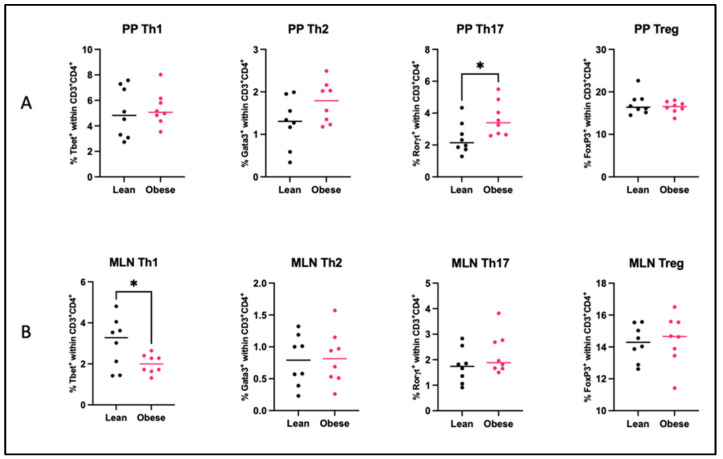
Percentages of T helper 1 (Th1) cells, T helper 2 (Th2) cells, T helper 17 (Th17) cells, and Regulatory T (Treg) cells in the (**A**) Peyer’s patches (PP) and (**B**) mesenteric lymph nodes (MLN) of obese and lean mice at day 12 of pregnancy. Data are displayed as individual values along with the median. Mann–Whitney U test. * *p* < 0.05). Lean mice: *n* = 8. Obese mice: *n* = 8.

**Figure 6 ijms-25-09076-f006:**
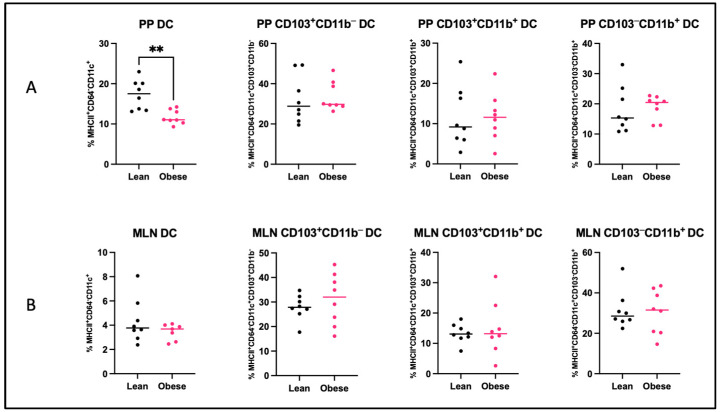
Percentages of dendritic cells (DC), CD103^+^CD11b^−^ dendritic cells, CD103^+^CD11b^+^ DCs, and CD103^−^CD11b^+^ dendritic cells in the (**A**) Peyer’s patches (PP) and (**B**) mesenteric lymph nodes (MLN) of obese and lean mice at day 12 of pregnancy. Data are displayed as individual values along with the median. Mann–Whitney U test. ** *p* < 0.01). Lean mice: *n* = 8. Obese mice: *n* = 7–8.

**Figure 7 ijms-25-09076-f007:**
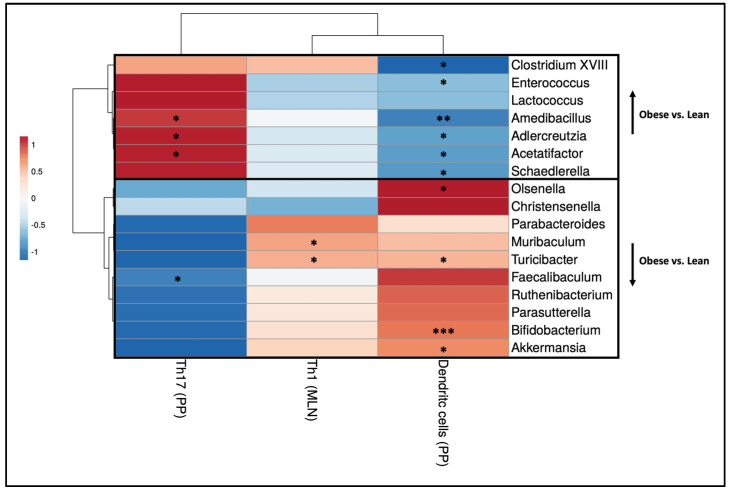
Heatmap displaying Spearman’s correlation coefficients resulting from individual correlations between intestinal immune cell populations (x-axis) and bacterial genera (y-axis) that exhibited significant differences between obese and lean mice at day 12 of pregnancy. Lean mice: *n* = 8. Obese mice: *n* = 8. * *p* < 0.05, ** *p* < 0.01, *** *p* < 0.001.

**Figure 8 ijms-25-09076-f008:**
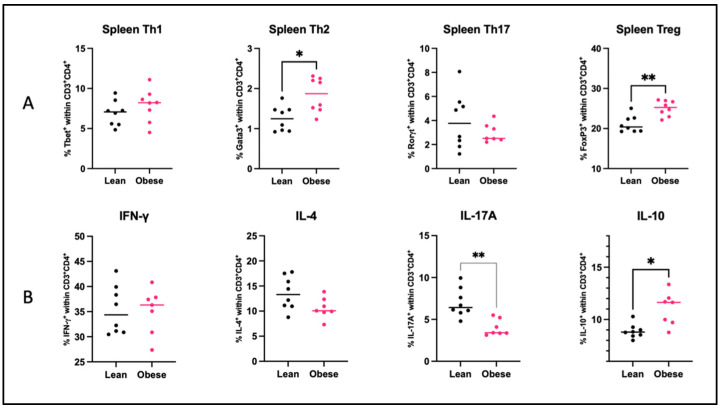
(**A**) Percentages of splenic T helper 1 cells (Th1), T helper 2 cells (Th2), T helper 17 (Th17) cells, and Regulatory T (Treg) cells of lean and obese mice at day 12 of pregnancy. (**B**) Percentages of IFN-γ, IL-4, IL-17A, and IL-10 producing splenic T helper cells (CD3^+^CD4^+^) upon ex vivo stimulation with PMA of lean and obese mice at day 12 of pregnancy. Data are displayed as individual values along with the median. Mann–Whitney U test. * *p* < 0.05, ** *p* < 0.01. Lean mice: *n* = 8. Obese mice: *n* = 7–8.

**Figure 9 ijms-25-09076-f009:**
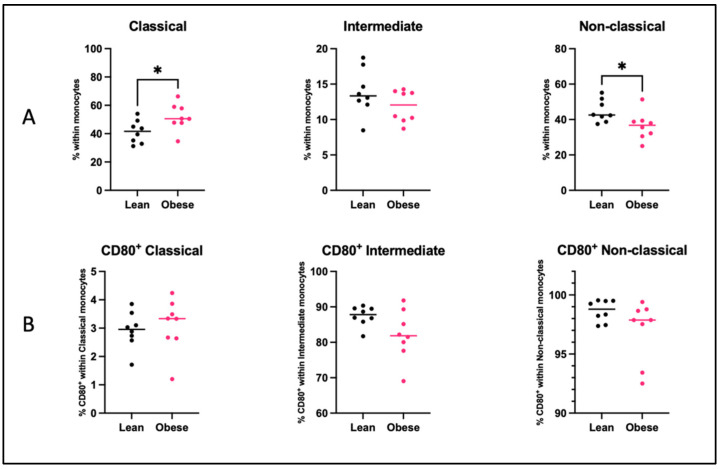
Percentages of (**A**) classical monocytes, intermediate monocytes, and non-classical monocytes and (**B**) CD80^+^ classical, intermediate, and non-classical monocytes in the blood of lean and obese mice at day 12 of pregnancy. Data are displayed as individual values along with the median. Mann–Whitney U test. * *p* < 0.05. Lean mice: *n* = 8. Obese mice: *n* = 8.

**Figure 10 ijms-25-09076-f010:**
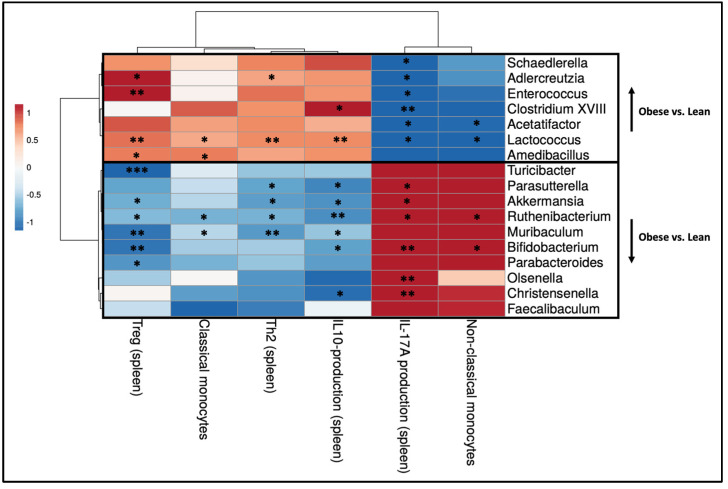
Heatmap displaying Spearman’s correlation coefficients resulting from individual correlations between peripheral immune cell populations (x-axis) and bacterial genera (y-axis) that exhibited significant differences between obese and lean mice at day 12 of pregnancy. Lean mice: *n* = 8. Obese mice: *n* = 8. * *p* < 0.05, ** *p* < 0.01, *** *p* < 0.001.

**Figure 11 ijms-25-09076-f011:**
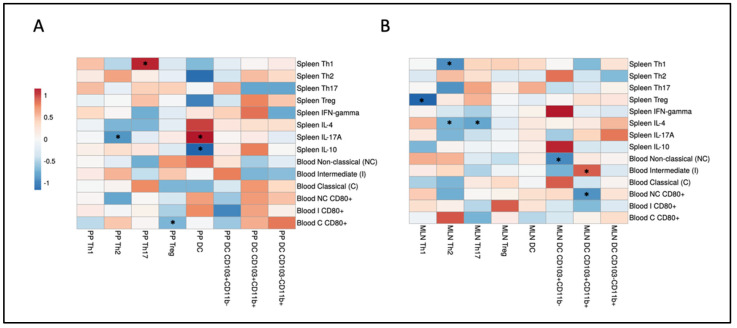
Correlations between intestinal and intestinal/peripheral immune cell populations. Heatmaps displaying Spearman’s correlation coefficients resulting from individual correlations of immune cell populations in (**A**) Peyer’s patches (PP; x-axis) and spleen/blood (y-axis) and (**B**) mesenteric lymph nodes (MLNs; x-axis) and spleen/blood (y-axis). Lean mice: *n* = 8. Obese mice: *n* = 8. * *p* < 0.05.

**Figure 12 ijms-25-09076-f012:**
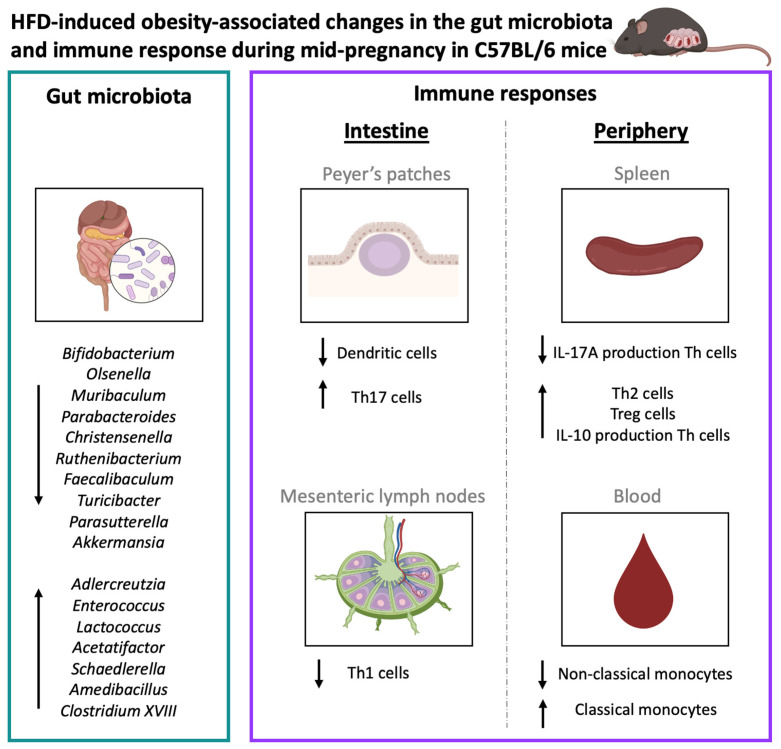
High-fat diet-induced maternal obesity-associated changes in the maternal gut microbiota and immune response in the intestine (Peyer’s patches and mesenteric lymph nodes) and the periphery (spleen and blood) during mid-pregnancy at gestational day 12 (GD12) in C57/BL6 mice. Upward arrows indicate an increase, while downward arrows indicate a decrease. Created with BioRender.com.

## Data Availability

The raw data supporting the conclusions of this article will be made available by the authors upon request.

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
