# Peer review of "Diet-Induced Obesity in Mice Affects the Maternal Gut Microbiota and Immune Response in Mid-Pregnancy"

_ijms, 2024, doi:10.3390/ijms25169076_

Round 1
Reviewer 1 Report
Comments and Suggestions for Authors
This article explores how maternal obesity affects mid-pregnant mice by impacting the gut microbiota and immune response, which is a novel research topic. The study found that compared to lean mice, maternal obesity affected the abundance of specific gut bacterial genera and disrupted maternal intestinal immune responses, subsequently altering the peripheral maternal immune responses during mid-pregnancy. However, there are also some issues with the article:
1. It is recommended that the authors include more gestational time points in future studies to assess the impact of obesity on different stages of pregnancy.
2. Although the study analyzed the composition of the gut microbiota in feces, it does not clearly indicate which specific bacteria changes caused the differences in gut and peripheral immune responses between obese and lean mice during pregnancy. It is recommended that further experiments delve deeper into this issue.
3. Why were fecal microbiota transplantation experiments not utilized to verify whether the differences in gut and peripheral immune responses between obese and lean mice during pregnancy were caused by the gut microbiota? It is suggested that such experiments be added in future research.
4.In Figure 2, what are the numerical values for the first principal component (PC1) on the X-axis and the second principal component (PC2) on the Y-axis? If available, please label them in detail.
5. Are Table A1 and Table S1 the same? The same question applies to Figure S1 and Figure A1.
6. Formatting issue at line 72.
7. In the manuscript, the 'P' in 'P < 0.01' should be uniformly in italics.
Reviewer 2 Report
Comments and Suggestions for Authors
The manuscript entitled „Diet-induced obesity in mice affects the maternal gut microbiota and immune response in mid-pregnancy" is very interesting and focuses on the influence of obesity on the outcome of pregnancy and, more importantly, on the correlation between obesity and gut microbiota during pregnancy. In the elegantly designed experiments, the Authors evaluated the impact of diet-induced obesity in pregnant mice on the profiles of immune-competent cells isolated from Peyer's patches and mesenteric lymph nodes. Moreover, they indicated the significant differences in gut microbiota between obese and lean pregnant mice. The results are well-discussed, indicating the correlation between obesity, the immune response required for the implementation of the embryos, and the course of pregnancy. The authors discussed the influence of the gut microbiota on the aspects of pregnancy in obese mice. However, they focused only on the major groups like Bifidobacterium and Akkermansia. The Authors should try to discuss the impact of microorganisms that belong to Adlercreutzia on the immune response in obese pregnant mice. These bacteria present anti-inflammatory features, and maybe they participate with Th17 cells in shifting the immune response at this stage of gestation in obese mice. The information regarding this group of bacteria is relatively scarce, but they should be discussed.
Minor remarks
The manuscript should be informative; please indicate what the "E" letter in the description of the stage of gestation stands for. One can guess that describes the stage of embryonic life, but it should be clearly stated.
Please increase the description of abscissa and ordinate in the graph; the current form is difficult to read.
Please indicate the total number of animals used in the studies.
Line 408: The authors stated that „Pregnancy outcomes were measured ….." Please confirm that it was determined during the E12 term.
The paragraph 4.4 is unnecessary. The information presented in this section needs to be clarified. Were the cells stained directly in the tissue, prepared sections, or cell suspension and then analyzed in the flow cytometry? Please rewrite this part or remove the information from this paragraph and include it in the following sections.
After revision, I recommend the manuscript for publication.
Reviewer 3 Report
Comments and Suggestions for Authors
The manuscript titled "Diet-induced obesity in mice affects the maternal gut microbiota and immune response in mid-pregnancy" aimed to investigate the associations between obesity-induced dysbiosis and immune parameters during mid-pregnancy in mice.
The manuscript is well-written and employs appropriate methodology for conducting the study and analyzing the data. Below are a few recommendations/questions/comments:
Methods:
1) Numerous studies indicate that dysbiosis can result from obesity, high-fat diets (HFD), or pregnancy. Why were fecal samples not collected on embryonic day 1 (E1) of pregnancy or prior to mating? It is possible that obese mice begin the study with a microbiota composition already differing from that of lean mice. It would be valuable to examine how both obesity and pregnancy impact the microbiome and how these changes evolve over time.
Results:
1) As a rule, references should not be cited in the results section of a manuscript. Remove references from lines: 92, 126, 157, 241.
2) Supplemental figure 1: Body weights of obese mice exceeded those of lean mice by 30% after 8 weeks of feeding. According to Supplemental Figure 2, mice from both groups began the study with significantly different body weights. Please add markers (*) indicating statistically significant differences between the groups at each time point on the growth curve shown in Supplemental Figure 1.
Discussion:
1) Add a paragraph about the limitations of the study. Please consider the limitations of correlational analysis and any other point that the authors would change if doing the study again.
